# Cohesin Mutations in Cancer: Emerging Therapeutic Targets

**DOI:** 10.3390/ijms22136788

**Published:** 2021-06-24

**Authors:** Jisha Antony, Chue Vin Chin, Julia A. Horsfield

**Affiliations:** 1Department of Pathology, Otago Medical School, University of Otago, Dunedin 9016, New Zealand; chuevin@gmail.com; 2Maurice Wilkins Centre for Molecular Biodiscovery, The University of Auckland, Auckland 1010, New Zealand; 3Genetics Otago Research Centre, University of Otago, Dunedin 9016, New Zealand

**Keywords:** cohesin, cancer, therapeutics, transcription, synthetic lethal

## Abstract

The cohesin complex is crucial for mediating sister chromatid cohesion and for hierarchal three-dimensional organization of the genome. Mutations in cohesin genes are present in a range of cancers. Extensive research over the last few years has shown that cohesin mutations are key events that contribute to neoplastic transformation. Cohesin is involved in a range of cellular processes; therefore, the impact of cohesin mutations in cancer is complex and can be cell context dependent. Candidate targets with therapeutic potential in cohesin mutant cells are emerging from functional studies. Here, we review emerging targets and pharmacological agents that have therapeutic potential in cohesin mutant cells.

## 1. Introduction

Genome sequencing of cancers has revealed mutations in new causative genes, including those in genes encoding subunits of the cohesin complex. Defects in cohesin function from mutation or amplifications has opened up a new area of cancer research to which several groups have contributed and provided insights into its impact and therapeutic potential. Here, we review underlying pathways that are dysregulated in cohesin mutant cancers and discuss some of the emerging targets that could have future therapeutic potential.

## 2. Cohesin Structure and Dynamics

Cohesin is a highly conserved ATPase complex in vertebrates best known for its canonical role in establishing sister chromatid cohesion during cell division [1,2,3]. The mitotic cohesin complex forms a ring-shaped structure that comprises four core subunits: RAD21, SMC3, SMC1A and STAG1/2 (Figure 1A) [2,4]. Chromatin association of cohesin is a dynamic process that is regulated by several cofactors (Figure 1). Cohesin is loaded onto chromatin through the action of NIPBL-MAU2 [5] and unloaded by association with WAPL-PDS5A/B [6,7]. These loading and unloading processes balance the amount of cohesin on chromatin for correct chromosome structure and function [7,8,9,10].

During DNA replication in S phase, sister chromatid cohesion is established via acetyltransferases ESCO1/ESCO2 that acetylate SMC3 (Figure 1B) [11,12,13,14], a process that requires the STAG2 subunit [15]. Acetylated SMC3 recruits Sororin to PDS5 and displaces WAPL to prevent unloading [16,17]. In prophase, most cohesin is unloaded following phosphorylation of STAG by Polo-like kinase 1 (PLK1), phosphorylation of Sororin by Aurora Kinase B and Cyclin-dependent kinase 1 (CDK1), which causes Sororin to dissociate from PDS5 and restores PDS5-WAPL [17,18,19,20]. However, centromeric cohesion is maintained by the counteracting activity of Shugoshin 1 (SGO1) and Protein Phosphatase 2A (PP2A) [21,22,23]. At anaphase, centromeric cohesin is removed via enzymatic cleavage of RAD21 by separase [24]. Unloaded cohesin complexes undergo deacetylation of the SMC3 subunit by HDAC8 and, then, can be reused [25] (cohesin dynamics reviewed in ref [26]).

In most vertebrates, the two mitotic complexes cohesin–STAG1 and cohesin–STAG2 are required to maintain sister chromatid cohesion at telomeres and centromeres, respectively [27,28]. STAG proteins form a crucial interface for the interaction of regulatory subunits and cofactors. Additional regulators that associate with the core cohesin complex have also been identified. BRCA2 modulates sister chromatid cohesion by limiting PDS5-mediated STAG association and regulates cohesin binding at DNA replication origins [29,30]. PDL1 can substitute for Sororin function in a context-dependent manner [31].

## 3. Cohesin Function

Cohesin is involved in a broad range of cellular functions. During mitosis, cohesin has roles in kinetochore bi-orientation [32], spindle assembly [33] and mitotic book-marking of transcription factors [34]. However, only a fraction (~13%) of cohesin is involved in these cell cycle functions. Most cellular cohesin functions in interphase [35].

In interphase, cohesin entraps DNA and, using its ATPase activity, extrudes DNA to form loops that are restricted by chromatin-bound CTCF (Figure 2) [3,36,37,38]. Cohesin-mediated DNA loops facilitate gene enhancer–promoter communication or form insulated hubs that prevent ectopic transcription [39,40,41]. Cohesin along with CTCF is crucial for hierarchal organization of DNA loops into topologically associated domains (TADs) [40,42,43,44]. TADs are further segregated into compartments or can even be part of larger compartments, representing active or inactive chromatin [7,37,40,41,45,46]. The function of cohesin regulatory proteins, NIPBL, WAPL and PDS5, is also crucial for accurate loop extrusion and chromatin organization [7,8,46,47]. RAD21 and NIPBL depletion strengthens compartmentalization, which led to the view that cohesin mediated loop extrusion counteracts epigenetic compartmentalization [40,41,46]. It has been suggested that such counteraction may be required to prevent the largescale spread of transcriptionally active or inactive states [40,41,46]. Cohesin also interacts with Polycomb group (PcG) complexes to influence organization of PcG-associated domains [48,49,50]. Cohesin can both facilitate as well as antagonize PcG (PRC1/PRC2)-mediated chromatin interactions, depending on the genomic site and cell context [48,49,50].

Cohesin STAG1 and STAG2 subunits have both redundant and non-redundant roles in chromatin organization and gene expression [48,51,52,53,54,55]. Both STAG variants co-localize with CTCF and have common and independent genomic binding sites. Cohesin–STAG1 largely functions to stabilize TAD boundaries and disrupt long range PRC1 interactions, which counteracts compartmentalization [48,54,56]. Cohesin–STAG2 mainly facilitates local enhancer–promoter interactions and long-range PRC1 interactions required for gene repression, all of which contribute to defining cell identity [48,54,55,56]. TADs are largely preserved upon STAG1 or STAG2 depletion, suggesting that they can compensate for each other to maintain chromatin architecture [48,54,56]. However, in the absence of STAG2, cohesin–STAG1 cannot compensate at a subset of STAG2-only sites found at key lineage defining genes [54,55].

Cohesin’s role in the hierarchal organization of chromatin explains much of its requirement for normal cellular function and development. Alterations in TADs and compartments upon cohesin depletion in different systems further support a crucial role for cohesin in genome organization. [7,41,43,44,46]. However, despite loss or alterations in TADs upon cohesin depletion, in several instances, only limited changes in gene expression have been observed [41,57]. Depletion of Nipbl in mouse liver cells resulted in equivalent loss of TADs in regions where gene expression was altered and in regions where expression remained unchanged [46]. A surprising recent study in *Drosophila* showed that in transcription-driven cell differentiation during gastrulation, 3-Dimensional (3D) chromatin conformation was relatively constant between tissues [58]. The study implies that 3D chromatin structure, rather than being instructive to cell type, instead acts as a scaffold for gene expression when enhancers become active. This interpretation is consistent with the finding that major transcriptional changes upon cohesin depletion are likely to occur when external signaling cues that induce transcription factor activity are applied to cells [57,59,60].

Cohesin has a crucial role in DNA replication (Figure 3). It interacts with DNA replication machinery and influences origin firing activity [61,62]. Cohesin also facilitates the progression of replication fork through hard-to-replicate regions and is required for the restart of replication stress-induced stalled fork [28,63]. Cohesin–STAG2 depletion in human telomerase reverse transcriptase (hTERT)-immortalized retinal pigmented epithelial cells impairs association with components of the replication machinery, reduces SMC3 acetylation and leads to replication fork collapse [15]. Cohesin removal at the replication fork by WAPL and PDS5 is required for fork protection and restart [64,65,66,67]. It was suggested that WAPL/PDS5-mediated cohesin removal causes remobilization or re-diversion of cohesin from the front to behind the fork and during replication stress this promotes recruitment of fork protection factors RAD51, BRCA2, WRNIP1 and inhibition of MRE11-associated degradation [65,67]. However, BRCA2 and RAD51 accumulation during replication stress is compromised by cohesin SMC1 depletion [67]. Based on this, it has been suggested that while dynamic cohesin removal is required for replication fork progression, the presence of cohesin is needed for fork protection [67]. 

Cohesin accumulates at DNA strand breaks and facilitates homology-directed DNA repair (Figure 3) [68,69,70,71]. Interestingly, separase also accumulates in interphase at sites of DNA breaks, and its function is also needed for homology directed repair [72]. This contrasting evidence suggests that there is dynamic cohesin turnover at DNA break sites, and that such cycling might be needed for cohesin recruitment [72]. Additionally, cohesin is also involved in the ATM/ATR-mediated DNA damage checkpoint response. Both SMC3 and SMC1 subunits are phosphorylated by the ATR/ATM kinase in response to ionizing radiation, which is required for enhanced mobilization of cohesin following DNA damage [73,74,75]. Cohesin–STAG2 was also shown to repress transcription at regions around double strand breaks [76]. Failure to repress transcription at DNA double strand breaks has the potential to lead to chromosomal translocation [76]. 

Cohesin also has a role in maintaining ribosomal DNA structure (rDNA), ribosomal RNA (rRNA) transcription and ribosome biogenesis [77,78,79]. Depletion or mutations in cohesin subunits or regulators impairs rRNA production and ribosome biogenesis [77,78,79]. The nucleolus is the hub of ribosome biogenesis, and aberrant nucleolar morphologies are observed in cohesin depleted models [78,80,81]. In yeast, during mitosis, cohesin was shown to be required for restricting chromatin contacts between rDNA repeats, and between rDNA and the rest of the genome [82]. Cohesin function in facilitating DNA replication appears to be essential for rRNA production and nucleolar integrity [79]. Depletion of the replication fork barrier mediator, *FOB1*, could partially rescue rRNA production and nucleolar defects in Eco1(ESCO2 human homolog) mutant yeast [79]. The highly transcribed repetitive rDNA regions are also prone to DNA breaks and unequal recombination, which can lead to chromosome instability. Cohesin, via its chromosome cohesion function, has a role in ensuring rDNA integrity [83,84]. Therefore, a combination of cohesin functions is involved in maintaining rDNA structure and function. 

## 4. Cohesin in Developmental Disorders

Complete loss of cohesin is incompatible with life, due to its crucial role in cell cycle. This means that cohesin mutations found in cells are largely haploinsufficient, except for STAG2 and SMC1A, which are on the X chromosome. STAG2 is compensated by STAG1. 

Haploinsufficient germline mutation in cohesin subunits or cohesin regulators causes a spectrum of multifactorial development disorders collectively known as the “cohesinopathies” [85]. Cornelia de Lange syndrome (CdLS) is the most prominent cohesinopathy and arises from heterozygous mutations, usually in the gene encoding cohesin regulator NIPBL, but sometimes in genes encoding the core mitotic-specific subunits of cohesin (RAD21, SMC3 and SMC1A) and the SMC3 deacetylase HDCA8 [85]. Clinically, CdLS is characterized by multisystem anomalies with cognitive deficits. The severity of the phenotype depends on which component of cohesin is mutated, with NIPBL mutations being most frequent (>65%) and accounting for the more severe clinical phenotypes [86,87]. Dysregulated gene expression is the major contributor to the pathology of CdLS. 

Germline mutations in STAG2 are classified under different umbrella of cohesinopathies, and these patients share some of the clinical phenotypes with CdLS [88,89]. Roberts syndrome (RBS) is a recessive cohesinopathy caused by homozygous mutation in ESCO2 [90]. Defects in sister chromatid cohesion and chromosome mis-segregation are present in RBS patients and may explain the underlying pathology of the syndrome [91]. Cohesin subunit mutations have also been identified in holoprosencephaly, a development disorder characterized by incomplete forebrain division [92]. A homozygous recessive SGO1 germline mutation causes the disorder chronic atrial and intestinal dysrhythmia, which affects the heart and gut [93]. Interestingly, in addition to loss of function, microduplications involving the *STAG2* gene have also been linked to a cohesinopathy with intellectual disability and behavior deficits [94]. 

## 5. Cohesin in Cancer 

Somatic mutations in cohesin genes and regulators are associated with several types of cancer [95,96,97] including glioblastoma (4–6%) [98,99], Ewing’s sarcoma (17–20%) [100,101], bladder (11–36%) [102,103,104,105,106] and myeloid neoplasms (13%) [107,108,109,110,111]. Mutations in genes encoding cohesin core subunits are largely mutually exclusive. No specific mutation hotspots have been identified. *STAG2* is the most frequently mutated subunit and is one of 12 genes to be mutated in four or more cancers [97,112]. In some cases of myeloid neoplasms and Ewing sarcoma, low expression of cohesin genes in the absence of mutations have been identified [100,109]. In addition to cohesin gene mutations, cohesin and CTCF binding sites are mutated in several cancers [113]. Cohesin mutation by itself does not lead to malignancy, but must co-occur with other mutations and co-operate with aberrant signaling events for cancer progression [100,101,106,111].

The clinical prognostic impact of cohesin mutation in cancers is not clear. In myeloid leukemias, cohesin mutation has been associated with both favorable [114], unfavorable [115] or of no significance to patient outcome [108]. According to a recent molecular classification of acute myeloid leukemia (AML), cohesin mutations are grouped in the chromatin-spliceosome group, which is associated with worse clinical outcomes [111]. Recently, a study reported that cohesin subunits and regulators are lowly expressed (from shallow to deep deletions) at high frequency (up to 75–95%) in a range of cancers, and this correlated with a significant decrease in patient survival [116].

Cohesin gene mutations can be truncating or missense [97]. Two groups have shown that not all tumor-derived cohesin core subunit mutations completely abrogate complex formation [117,118]. Surprisingly, subunit missense mutations, and also a subset of nonsense mutations, retained the ability of the subunit to interact with other complex members [117,118]. Using nuclear lysates from isogenic glioblastoma lines with intact or mutant STAG2, Kim et al. found that a truncating STAG2 mutation also reduced interaction of the core complex with WAPL/PDS5A/5B [117]. Furthermore, Rittenhouse et al. examined the impact of tumor-derived cohesin mutations by introducing them into mouse embryonic stem (ES) cells via CRISPR-Cas9 editing and found that different variants of SMC1A had different effects on cell growth, differentiation and gene expression [119].

Cohesin deficiency in cell models also triggers chromosome and cell cycle defects [117,120,121]. It was initially thought that cohesin mutations in cancers would be associated with aneuploidy. Guo et al. identified increased incidence of chromosome copy number variation in their cohort of cohesin mutant bladder cancers [103]. However, other studies did not find a link between cohesin mutation and aneuploidy in bladder cancer [102,105]. Similarly, no increased incidence of aneuploidy was observed in cohesin mutant Ewing sarcoma [100] or myeloid neoplasms [107,109]. Interestingly, Kim et al., to identify a link between cohesin mutation and aneuploidy, created HCT116 cell lines to contain tumor-derived STAG2 mutation [117]. Nonsense STAG2 mutations adversely affected chromatid cohesion and led to lagging chromosomes, while no such effects were triggered by missense mutations [117]. In spite of chromosome segregation defects, modal increase in chromosome number was only detected in one STAG2 mutant line [117]. Another study found that while one missense STAG2 mutation reduced STAG2′s ability to repress transcription at double strand breaks, another missense mutation had no effect on this function [76]. Reduced expression of cohesin was also found to increase chromosome instability by increasing micronuclear formation and nuclear size [116]. These studies show that while in some instances cohesin mutations can lead to chromosome instability, this does not necessarily result in aneuploidy.

Evidence from several studies suggests that the main mechanisms by which cohesin mutations contribute to cancer is by disrupting genome organization and transcription [54,55,59,60,81,118,122,123,124,125,126,127,128,129]. Aberrant DNA looping and dysregulation of key lineage transcription factors involved in cellular identity or homeostasis are observed upon cohesin depletion. Abnormal cellular plasticity is central to malignant transformation, and cohesin status affects the balance between self-renewal and differentiation. Direct evidence linking cohesin mutations to aberrant transcription that causes neoplastic transformation comes from studies carried out in hematopoietic stem cells (HSCs). In HSCs, cohesin insufficiency enhances self-renewal and impairs differentiation [55,57,81,118,128,130,131,132,133]. Cohesin insufficiency alters chromatin accessibility and/or DNA looping and causes misexpression of HSC-specific genes such as *HOX* genes and dysregulation of hematopoietic transcription factors such as *RUNX1*, *ERG*, *GATA1*, *Ebf1* and *Etv6* or genes involved in inflammation. In combination, these factors reinforce the HSC state and impair differentiation. Knockdown of *RUNX1*, *ERG* and *GATA1* can reduce self-renewal of cohesin mutant HSCs [128] and restoration of *Ebf1* expression rescued B cell differentiation of cohesin STAG2-depleted HSCs [55], which suggests that altered HSC homoeostasis is due to transcription dysregulation.

In mouse embryonic stem (ES) cells, depletion of cohesin subunits or Nipbl reduced enhancer–promoter interaction at pluripotency genes, which decreased their expression and consequently resulted in loss of ES cell state [134,135]. A similar decrease in pluripotency was also observed in mouse ES cells deficient for Stag1/2, or with tumor-derived cohesin mutations in Stag2 or Smc1a [119]. RNAi knockdown of SMC1A or SMC3 in human epidermal progenitors cells impairs chromatin accessibility, downregulates genes associated with self-renewal and upregulates differentiation genes [136]. Depletion of Rad21 and Nipbl in Drosophila intestinal stem cells activates differentiation-associated transcription that could be rescued by overexpression of intestinal stem cell master transcription factor, escargot [137]. In epithelial cancer cells, RAD21 depletion alters chromosome interactions around *TGFB1* and *ITGA5*, leading to their activation and epithelial to mesenchymal transition [127]. STAG2 knockout in Ewing sarcoma cell lines was found to downregulate EWSR1-FLI1-anchored chromatin interactions and enhance the migratory potential of these cells [138]. Together these studies show that cohesin insufficiency can result in diverse aberrant cellular phenotypes.

## 6. Overexpression of Cohesin in Cancers

Cohesin genes are also overexpressed from copy number gains in several cancers including breast [139], colorectal [140] and lung cancers [141]. Among the cohesin genes, it is *RAD21* that is most frequently amplified in cancers. Cohesin *RAD21* overexpression in *HER2* mutant breast cancers and *KRAS* mutant colorectal cancers is associated with poor prognosis and resistance to treatment [139,142]. *RAD21* overexpression was assumed to be a passenger effect due its location within the frequently amplified genomic locus 8q24 that also harbors the oncogene *c-MYC*. However, there is evidence that overexpressed *RAD21* itself contributes to the pathology of cancer. RAD21 is required for *c-MYC* transcription [125,129]. In breast cancer cells, RAD21 is required for expression of a subset of estrogen induced genes [60,125]. Adenomatous polyposis mutation-driven Wnt activation was found to elevate RAD21 levels [143]. Approximately 50% of Ewing sarcoma with EWSR1-FLI1 harbor trisomy 8. Gain of *RAD21* copy number due to trisomy 8 was found to reduce replicative stress and support the growth of Ewing sarcoma cells harboring EWSR1-FLI1, and this effect was independent of *STAG2* mutation status [144]. Thus, it appears that dosage of cohesin needs to be exquisitely maintained. Both cohesin gain (for example, by gene copy number increase) or reduction (for example, by gene deletion or mutation) can contribute to neoplastic transformation. In this review, we focus on cohesin mutation or loss in cancers because these represent the majority of cancer-associated cohesin lesions.

## 7. Therapeutic Targeting in Cohesin Mutant Cancers

Potential avenues for targeting cohesin mutant cells have emerged from both targeted and genome-wide approaches. Targeted approaches include studies carried out to elucidate the function of subunits or identify enzymatic regulators of the complex, or to identify subunit interaction sites [25,145,146,147,148]. Functional studies on the consequences of cohesin depletion have revealed transcription factors or pathways that can be modulated [55,59,128]. Direct modulation of transcription factors is challenging; therefore, pharmacological agents [59] that target the epigenetic regulation of transcription are more frequently used.

Genome-wide approaches have been based on identifying synthetic lethal interactions with cohesin depletion, i.e., genes that when mutated, compromise the viability of cohesin-deficient cells. The first synthetic lethal genetic screen was conducted by McLellan et al. in *S. cerevisiae* with mutations in genes encoding cohesin subunits (smc1, scc1) and NIPBL (scc2) [149]. More recently, dropout CRISPR-Cas9 screens in isogenic cancer cells have been conducted to identify genetic synthetic lethal targets with STAG2 mutations [148,150]. Genome-wide high-throughput drug screening allows direct identification of druggable targets. We used isogenic MCF10A breast epithelial cell lines with deletion mutations in cohesin subunit genes *RAD21*, *SMC3* and *STAG2* in a synthetic lethal screen with 3009 FDA approved drugs [80]. This screen identified classes of compounds that commonly inhibited all three cohesin mutant lines and as well as compounds that were subunit-specific.

We review below emerging strategies of therapeutic targeting in cohesin mutant cells. We classified these strategies into three categories: direct targeting of cohesin components or its regulators (Figure 1); targeting underlying dysregulated transcription or signaling events (Figure 2); targeting DNA damage repair (Figure 3). A summary of agents with therapeutic potential in cohesin mutant cancers is presented in Table 1.

### 7.1. Targeting Cohesin Complex Assembly

Cohesin subunits: The mutual exclusivity of cohesin subunit mutations in cancer and developmental disease implies that one-subunit mutation is sufficient to compromise cohesin complex function. However, evidence suggests that not all cohesin subunit functions are completely overlapping. In Drosophila, independent cohesin subunits are differentially enriched at genomic sites. Enhancers are mainly associated with the STAG subunit (SA), while Rad21 and Nipbl are found at promoters [165]. These differential associations have implications for the formation of the full complex, including its recruitment and function. Whether the identity of the mutated subunit influences how cohesin-mutant cells would respond to therapeutics is currently an under-investigated area. We recently identified 18 synthetic lethal compounds that inhibited growth of all three cohesin mutant (*RAD21*, *SMC3* and *STAG2*) cells compared to parental MCF10A [80]. However, different cohesin mutant lines also had independent sensitivities to compounds. In individual analyses, the *STAG2* mutant line had the largest number of synthetic lethal hits (n = 157), while the *RAD21* mutant line was sensitive to the fewest compounds (n = 27) [80]. “Hits” from the *STAG2* mutant line spanned more diverse classes and included compounds in immunological and G protein-coupled receptor categories that were not identified among *RAD21* and *SMC3* hits [80]. Our findings suggest that, depending on the subunit mutated, there is likely to be differences in sensitivity to compounds targeting cohesin-dependent pathways.

STAG1: Synthetic lethal interaction between STAG1/2 paralogs has been identified in several cancer cells (Figure 1A) [15,145,147,148,151]. STAG1 depletion in cells with normal cohesin or intact STAG2 has no impact on cell proliferation [15,147]. However, inactivation of STAG1 leads to cell death in *STAG2* mutant cells [145,147,148,151]. The mechanism underlying cell death is a loss of compensation for cell cycle function of cohesin–STAG2, leading to chromosome mis-segregation [166]. Depletion of the other core subunits RAD21 and SMC3 reduced growth in both parental and STAG2 mutant cells similarly. These findings suggest a strong redundancy of STAG1 and STAG2 function in supporting cohesin’s role in mitosis [15,148].

Van der Lelij et al. used an auxin degron system to demonstrate that selective degradation of STAG1 reduced the viability of *STAG2* mutated cells without affecting *STAG2* wild type cells. They suggested that in a therapeutic setting, the emerging technology proteolysis targeting chimera (PROTAC) could be used for selective targeting of STAG1 [148]. The group further showed that D797 residue of STAG1 was crucial for the interaction of STAG1 with RAD21 [148]. Blocking this interaction was highlighted as another potential avenue for therapeutic targeting of STAG1 in *STAG2* mutant cancers [148].

SMC3: Kang et al. had shown that Glycyrrhizic acid (GA), a derivative of licorice, inhibited growth of lymphocytes infected with Kaposi’s sarcoma-associated herpesvirus by blocking RNA polymerase II enrichment at cohesin–CTCF sites, blocking SMC3 acetylation and its interaction with RAD21 [152]. Interestingly, GA has been shown in vitro to reduce viability of cancer cells [167].

HDAC8: The HDAC8 enzyme is necessary for deacetylation and recycling of cohesin’s SMC3 subunit [150]. PCI-30451 was identified as a potent inhibitor of HDAC8, with potential to prevent cohesin recycling [150,168]. In breast cancer cells, we found that PCI-30451 caused accumulation of acetylated-SMC3 and disrupted cell cycle progression [146]. However, in contrast to RAD21 or SMC3 knockdown, PCI-30451 treatment had no impact on cohesin regulation of gene transcription [146] and appears to interfere with cohesin’s role in the cell cycle. PCI-30451-mediated HDAC8 inhibition induces cell death in a range of cancers including AML [169]. It is not clear that HDAC8 inhibition acts through blocking SMC3 recycling alone: HDAC8 deacetylates many substrates including p53 and estrogen-related receptor alpha [170]. This suggests that HDAC8 inhibition would have diverse molecular consequences depending on the cell type.

Separase: The separase enzyme cleaves RAD21 to ensure sister chromatid separation at the metaphase–anaphase transition. Sepin-1 has been identified as a potent inhibitor of separase [154]. Separase overexpression is reported in 60% of breast cancers, and induction of separase is associated with aneuploidy [153,154]. Cancer cells that overexpress separase display more sensitivity to Sepin-1 [153,154,155]. Toxicology studies on Sepin-1 have so far been favorable [171]. Interestingly, separase is also recruited in interphase to DNA double-strand breaks where it cleaves RAD21 to remove cohesin and enable homology mediated repair [72,172]. Further studies in cohesin-mutant models would be valuable to determine whether separase inhibition could sensitize cohesin-deficient cancers.

Other cell cycle regulators: Inhibitors of cohesin regulators Aurora kinase B, polo-like-kinase 1, cyclin-dependent kinase 1 induce cell death in cancer and some others are currently in clinical trials [173]. These regulators have not emerged as differentially sensitive genetic hits in genome-wide CRISPR-Cas9 screens of cohesin mutant cell lines [148,150], probably because their mutation would result in similar genetic lethality in both normal and cohesin-deficient cells. However, cohesin-deficient cells in our synthetic lethal drug screen had increased sensitivity to the Aurora kinase B inhibitors, MK-8745 and ZM 44743 and cyclin-dependent kinase inhibitor, P276-00 [80], reflecting their cooperation in mitosis.

### 7.2. Modulating Transcription Using Inhibitors to Epigenetic Targets

Inhibition of transcription factors by pharmacological agents that target the epigenetic regulation of transcription is a heavily investigated area. In cohesin mutant cancers, targeting of DNA methylation or histone readers and modifications has emerging therapeutic potential (Figure 2).

DNA hypomethylating agents: Cohesin and CTCF preferentially co-bind to DNA hypomethylated regions, and CTCF binding is displaced by DNA hypermethylation [174]. DNA hypomethylation was shown to enable CTCF recruitment of cohesin to form loops that alter mRNA alternative cleavage and polyadenylation [175]. Cancers with hypermethylated DNA that also lose cohesin function could synergistically compromise cohesin’s ability to bind DNA and organize the genome. DNA hypomethylating agents (HMA) that inhibit DNA methyltransferases are already used to treat myeloid cancers, but with varying response rates [176]. Rationalized use of HMA could be revisited based on the underlying mutational status of the cancer. Thota et al. found that myeloid dysplasia patients with *STAG2* or *RAD21* mutations had a significantly better response to the HMA, decitabine and azacytidine than patients without cohesin mutations [109]. Subsequently, Tothova et al. found that adult CD34+ cells heterozygous for SMC3 mutation were more sensitive to azacytidine treatment compared to isogenic controls [156]. In cohesin mutant cancer cells, HMA could theoretically rescue loss of cohesin function by stabilizing the remaining cohesin on chromatin. In support of this idea, treatment of an in vitro model of IDH gliomas with azacytidine enhanced CTCF binding and partially restored genome insulation [177]. Furthermore, it is possible that stabilization of limited cohesin reduces cohesin recycling, which ultimately leads to loss of cell viability. Further investigation is required to understand the mechanisms behind why HMA are effective in cohesin mutant cancers.

Bromodomain and extra-terminal (BET) protein inhibitors: BET inhibitors are prominent epigenetic-modulating drugs that are in clinical trials. BET proteins bind to active acetylated histone residues, leading to activation of RNA polymerase II recruitment and gene activation [178,179]. BET inhibitors were developed to combat cancers with oncogenic overexpression of BET proteins (BRD2/3/4). Interestingly, *BRD4* germline mutations were categorized into the CdLS spectrum due to an overlap in clinical phenotype with *NIPBL* mutation [180]. BET proteins are also enriched along with cohesin at active enhancers [165,181,182], which lose insulation upon cohesin depletion or mutation [41]. The BET inhibitor JQ1 was shown to reduce cohesin–RAD21 binding from some CTCF sites [183]. We found that *STAG2* mutation in K562 leukemia cells increased chromatin accessibility at BRD4 binding sites at the *RUNX1* and *ERG* genes [59]. Treatment with JQ1 prevented inducible transcription of *RUNX1* and *ERG* in *STAG2* mutant cells [59]. *RUNX1* dysregulation can arise from aberrant activity of its eR1 enhancer [124]. Mill et al. showed that *RUNX1* expression in leukemia cells could be suppressed similarly by deletion of eR1, or by BET inhibitors, or BET protein degradation (BET PROTAC) [184]. BET inhibitors I-BET-762 and RVX-208 were also among the top hit compounds to which cohesin mutant cells displayed increased sensitivity in our isogenic MCF10A synthetic lethal drug screen [80]. These findings suggest that BET inhibitors have the potential to reduce aberrant transcription and growth of cohesin mutation cells.

Histone modifiers: Targeting enzymes involved in histone post translation modifications represents another strategy to interfere with cancer associated transcriptional dysregulation. In HSCs, cohesin RAD21 depletion was found to reduce the PcG-associated repressive H3K27me3 mark and cause de-repression of PcG target genes, particularly at leukemia-associated *HoxA7* and *HoxA9*, consequently enhancing self-renewal [131]. Upregulation of *HOXA* genes with concomitant decrease in H3K27me3 and increase in activating H3K27ac was also observed in *STAG2* mutant OCI-AML3 leukemia cells [160]. *HOXA9* expression is associated with aggressive leukemias. DOT1L, a histone methyltransferase, is an epigenetic reader that is recruited to genomic areas devoid of H3K27 methylation, where it generates the active H3K79me2 mark [157]. DOT1L inhibitors are recognized as potential therapeutics for MLL leukemias, which also have high *HOXA9* expression. Recently, Heimbruch et al. showed that cohesin depleted HSCs have increased global H3K79me2, including at the *HoxA9* locus. Pharmacological inhibition of DOTIL in cohesin mutant HSCs reduced H3K79me2, leading to downregulation of *HoxA9/A7*, suppression of self-renewal and re-activation of the differentiation transcription program [157]. The findings from this study have added DOT1L inhibition as a potential therapeutic in cohesin mutant cancers. Given cohesin’s interaction with PcG, further studies are needed to determine whether the H3K27me3 mark or PcG members could be modulated for therapeutics in cohesin mutant cells.

### 7.3. Targeting Signaling Pathways

Several pathways that alter proliferation and cellular identity are dysregulated in cohesin-deficient cells, and some have emerged with potential for therapeutic targeting (Figure 2).

Wnt signaling: Dysregulation of Wnt signaling as an underlying event in cohesin depleted cells has emerged from several studies [80,158,159,185]. We found that cohesin mutant MCF10A cells showed increased sensitivity to the glycogen synthase kinase 3 (GSK3) inhibitor LY209031, which stimulates Wnt signaling. One potential cause of Wnt sensitivity in cohesin-deficient cells could be due to enhanced β-catenin stabilization [80]. Increased sensitivity to Wnt stimulation was also observed in cohesin *STAG2* mutant leukemia cells, HCT116 colorectal cells that overexpress mutant SMC1A and in *rad21* and *stag2b* mutant zebrafish models [80]. Wnt-responsive transcription was also greatly sensitized in *STAG2* mutant leukemia cells [80]. It is unclear how β-catenin is stabilized in cohesin mutant cells. Factors that alter cellular pH influence β-catenin stabilization [186], and we speculate that cohesin insufficiency-associated oxidative stress, nucleolar fragmentation and compromised ribosome biogenesis might alter pH by altering cell metabolism [77,187]. Wnt agonism in cancer is controversial given that Wnt signaling promotes growth. Hyperactive Wnt signaling leading to increased frequency of myeloid progenitors was shown in Nipbl-depleted zebrafish, and in this study, treatment with the Wnt inhibitor indomethacin ameliorated this phenotype [159]. Lithium, another GSK3 inhibitor, rescued cell proliferation defects in a Drosophila CdLS model and in lymphoblastoid cell lines from CdLS patients [158]. Lithium and other GSK3 inhibitors have been shown be effective inhibitors in several cancers; however, further studies will be required to understand how Wnt agonism could be of therapeutic value in cohesin mutant cells.

Metabolism: Consistent with cohesin involvement in rDNA transcription, ribosome biogenesis and regulation of genes in PI3K-mTOR pathway [60,77,188], cohesin mutant cells have increased sensitivity to drugs targeting components of these growth signaling pathways in some experimental models. Cohesin mutant MCF10A cells showed increased sensitivity to inhibitors of mTOR (WAY-600 and AZD2014), AKT (Ipatasertib), BRAF (Dabrafenib), p38 MAPK pathway (VX-702) and VEGFR-3–tyrosine kinase (SAR131675) [80]. Furthermore, *STAG2* mutant OCI-AML3 cells are differentially sensitive to MEK inhibitors (Selumetinib and Trametinib), owing to downregulation of genes in the MAPK signaling pathway [160]. In contrast, *STAG2* or *STAG3* mutation in melanoma conferred resistance to BRAF inhibition owing to reactivation of MEK-MAPK (ERK) signaling, which suggests that in a melanoma context, *STAG2* mutation status could inform therapeutics [189]. No enhanced sensitivity to inhibitors of mTOR (Everolimus) and tyrosine kinase (Imanitab, Sorafenib) was observed in *STAG2* mutant glioblastoma, Ewing sarcoma and hTERT-positive retinal pigmented epithelial cells in Mondel et al.’s study [15]. These findings highlight a cell type dependency for sensitivity, and also that determining the mutational landscape of the cancer will be necessary to determine whether drugs targeting these key growth pathways would be beneficial in the different cohesin mutant cancers.

Inflammation: Cohesin is involved in the regulation of inflammatory genes and ensuring appropriate cellular response to inflammatory cues; for example, genes associated with inflammation are downregulated in cohesin mutant AML [57]. Cuartero et al. showed that RAD21-deficient macrophages fail to induce expression of inflammatory genes, including interferon response genes, upon stimulation with lipopolysaccharide (LPS) [57]. RAD21-depleted HSCs not only have reduced expression of inflammatory genes (interferon γ and NF-κB signaling) but also fail to differentiate in response to LPS pro-inflammatory cues [57,133]. Interestingly, Cuartero et al. also showed that interferon γ/β priming of RAD21-depleted macrophages prior to LPS assault could rescue enhancer activity and expression of inducible genes [57]. Thus, in HSCs and myeloid cells, cohesin is required for inflammatory gene expression.

The inflammation-associated role of cohesin in HSCs can also become a double-edged sword during aging and chronic inflammation [133]. Chronic inflammation or aging can exhaust the HSC pool. To counteract this, HSCs downregulate cohesin or the inflammatory transcription factor NF-κB [133]. This in turn selects for a pool of HSCs with reduced cohesin, or with cohesin mutation, that has enhanced self-renewal. These cells skew towards the myeloid lineage and are resistant to differentiation [133]. Therefore, inflammatory signals could drive clonal expansion of cohesin mutant cells in leukemia, such that suppressing inflammation (for example by using p38 inhibitors) might be therapeutically beneficial [190].

In contrast to HSCs, *STAG2* deletion in HT-29 human intestinal epithelial cells causes DNA damage and genome instability leading to excessive interferon production via the cGas-Sting pathway [191]. Cohesin depletion in HSCs is not associated with DNA damage or genome instability, which might account for this difference. In HeLa cells, cohesin depletion during interferon γ priming leads to an unconstrained interferon-associated transcriptional memory response [192]. Therefore, cohesin is required for the inflammatory response, but it can also modulate the magnitude of inflammatory gene expression. Research to date highlights that inflammatory signaling and sensitivity to inflammatory cues could vary between cohesin mutant cancers according to underlying differences in a DNA damage state.

Immune checkpoint: Programmed death 1 (PD1) receptor and its ligand programmed death ligand 1 (PDL1) are key immune checkpoint regulators that are targeted in cancer immunotherapy. Interestingly PDL1, but not PD1, is highly expressed in triple-negative breast cancer (TNBC). PDL1 expression negatively correlates with Sororin expression in TNBCs [31]. A subset of PDL1 was shown to be localized to the nucleus, where it functions to compensate for Sororin’s function in establishing sister chromatid cohesion [31]. This nuclear function of PDL1 was independent of interaction with PD1. Another study found that nuclear PDL1 interacts with cohesin–STAG1 in HeLa cells [161]. Inhibition of PDL1 disrupted cohesin–STAG1 and led to telomere dysfunction and chromosome segregation errors [161]. These studies provide insights into a heretofore unknown role of nuclear PDL1, which would likely aid in the use of PD1/PDL1 therapies for cancers. From research to date, it appears that PDL1 interacts primarily with the cell cycle function of cohesin; hence, combining anti-PDL1 therapy with other cohesin regulator inhibitors could be useful in targeting cohesin mutant cancers.

### 7.4. DNA-Damaging Agents

Cohesin-deficient cancers cells have increased sensitivity to ionizating radiation, owing to cohesin’s role in DNA double-strand break repair [15,193]. Therefore, using DNA-damaging agents to further block the compromised DNA repair pathway of cohesin mutant cells has potential for selective inhibition of these cells (Figure 3).

Poly (ADP-ribose) polymerase (PARP) inhibition: PARPs are enzymes that are recruited to sites of DNA double-strand breaks. PARP inhibition resulting in synthetic lethal killing was demonstrated in BRCA1/2 cancers that have compromised DNA repair pathways [194,195]. McLellan et al., using a yeast model, identified synthetic lethality between genes involved in replication fork dynamics, including PARP enzymes, with SMC1 mutation. They then showed that pharmacologic inhibition of PARP (benzamide and olaparib) induced synthetic lethal killing in cohesin-depleted colon neoplastic cell lines with intrinsic high or low PARP activity [149]. A similar increased sensitivity to PARP inhibition was observed in STAG2-depleted glioblastoma [15,163], Ewing sarcoma [15], hTERT-positive retinal pigmented epithelial cells [15] and in certain STAG2 mutant leukemia cells [150]. Depletion of SMC1 in triple-negative breast cancer cells increased sensitivity to PARP inhibitor ABT-888 [196]. Increased sensitivity to olaparib was also demonstrated with PDS5B (APRIN) depletion in breast cancer cells and in a zebrafish xenograft model [29,162]. Enhanced sensitivity to PARP inhibition using talazoparib was also demonstrated in a *Stag2/Tet2* mutant myeloid neoplasm mouse model [150].

Interestingly, Liu et al. reported that combined STAG1 knockdown and PARP inhibition was more effective at further reducing growth of STAG2 mutant Ewing sarcoma and bladder cancer cells [151]. We observed increased nuclear fragmentation and accumulation of γH2AX in cohesin mutant isogenic MCF10A cells upon treatment with the DNA-damaging agent actinomycin D [80], suggesting that cohesin mutant cells are more susceptible to induced DNA damage. PARP inhibition with olaparib only resulted in only a mild to moderate decrease in the growth of cohesin-deficient cells compared to parental MCF10A cells, with *STAG2* mutant cells being the most susceptible [80]. This suggests that cell type or the selection of co-existing mutations could be important for effective PARP inhibition in cohesin mutant cells.

PARP uses NAD for its enzymatic activity to tether poly(ADP-ribose) to itself or to its targets [197]. Current PARP inhibitors compete with NAD+ for binding at the active site of PARP enzymes [197]. Low intracellular NAD+ reduces PARP activity [164]. Nampt is the rate limiting enzyme involved in the synthesis of NAD+ [164]. Interestingly, Nampt inhibition reduces intracellular NAD+ and results in DNA hypermethylation of the *BDNF* gene, encoding brain-derived neurotrophic factor. Hypermethylation was accompanied by the release of cohesin/CTCF binding from *BDNF* regulatory sites and reduced *BDNF* expression [164]. A combination of Nampt and PARP inhibition has been shown to have a synergistic effect in arresting tumor growth [198]. Given the link between cellular NAD+ levels, cohesin binding and PARP activity, combinatorial Nampt/PARP inhibition is an avenue that could be explored for optimizing PARP inhibition-based therapeutics in cohesin mutant cells.

Other DNA-damaging agents: Targeted knockdown experiments and genome wide CRISPR-Cas9 screen analyses in isogenic cohesin mutant cell lines have identified synthetic lethality with genetic depletion of several other genes required for DNA repair and replication (such as ATR, RAD51, BRCA1, replication protein A2, POLD3) [15,150]. Several well-known chemotherapeutic DNA-damaging agents have been trialed in cohesin mutant cells. Knockdown of RAD21 in MDA-MB-231 breast cancer cells increased sensitivity to DNA-damaging agents cyclophosphamide, 5-fluorouracil and etoposide [139]. STAG2 mutant glioblastoma, Ewing sarcoma and hTERT-positive retinal pigmented epithelial cells are sensitive to several DNA alkylating agents (cyclophosphamide, gemcitabine and temozolomide), ATR kinase inhibitors (VX-970, AZD6738), topoisomerase poisons (doxorubicin, etoposide and topotecan) [15].

In estrogen receptor negative breast cancer, low PDS5B (APRIN) levels were shown to be associated with longer disease-free survival following adjuvant chemotherapy with TOP2 poison anthracycline [162]. Topoisomerases break and religate DNA to relieve torsional stress created by processes such as replication or loop extrusion. Topoisomerase poisons stabilize TOP I or II enzymes on the DNA, increasing the frequency of DNA breaks, causing them to become harder to repair and triggering apoptosis. However, persistent DNA breaks, when left unresolved can also lead to oncogenic chromosomal translocation [199,200,201,202]. Etoposide treatment is associated with therapy related AML [199,200,201,202]. Interestingly, etoposide-associated TOP2B-mediated DNA breaks occur more frequently at the base of cohesin/CTCF-anchored loops [199,200,201,202]. Cohesin is also required for TOP2B binding and activity [199,200,201]. Further studies are needed to determine the mechanism of topoisomerase inhibitor response in cohesin mutant cells.

## 8. Conclusions

The cellular consequences of cohesin dysregulation appear to depend on dosage of cohesin levels, subunit identity, mutation type, co-occurring mutations and cell type. While several potential therapeutic targets have emerged for cohesin mutant cells, further mechanistic studies in combinatorial mutant models are still needed. Understanding the mechanisms underlying increased sensitivity of cohesin-deficient cells to agents like HMA or BET inhibitors, or how combination therapeutics could be employed, might lead to tailored therapeutics for cohesin mutant cancers.

## Figures and Tables

**Figure 1 ijms-22-06788-f001:**
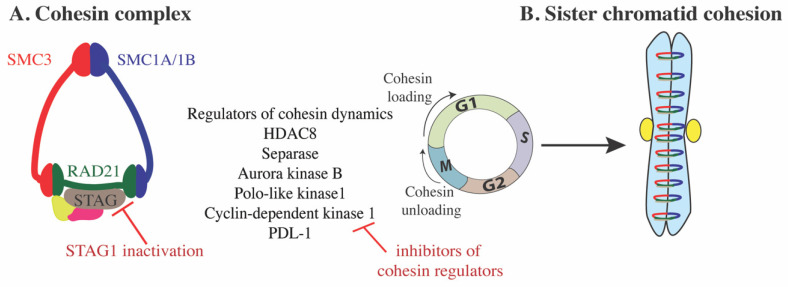
(**A**) Schematic of the cohesin complex. Cohesin loading and unloading onto chromosomes during cell cycle is dynamic and involves a number of regulators. STAG2 mutations are the most frequent, and inhibition of STAG1 subunit is synthetically lethal with STAG2 mutations. Listed are the cohesin regulators that can be pharmacologically inhibited and their inhibition mainly disrupts cell cycle progression. (**B**) Cohesin subunit SMC3 acetylation during S phase ensures sister chromatid cohesion.

**Figure 2 ijms-22-06788-f002:**
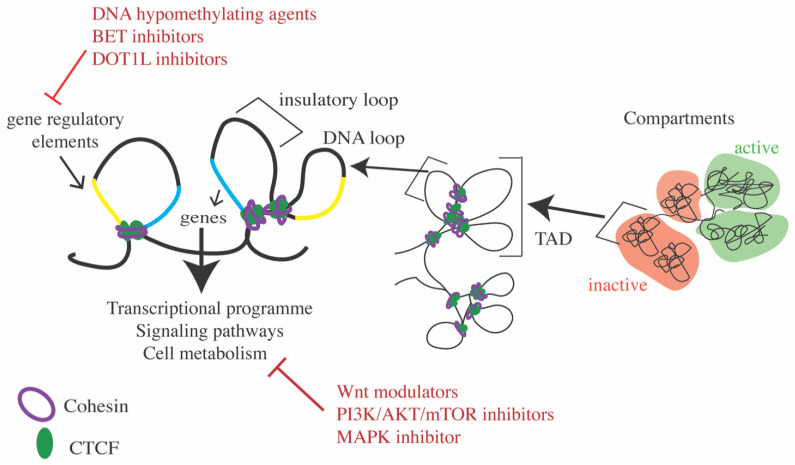
Cohesin’s role in the hierarchical 3-dimensional organization of the genome. Cohesin association with DNA during interphase is required for formation of DNA loops and organization into TADs. DNA loops allow genes to either connect to their regulatory elements (enhancers) or insulate them from ectopic connections. TADs based on transcription and epigenetic modifications segregate into active and inactive compartments. Cohesin mutation can result in aberrant DNA loops, which leads to transcriptional dysregulation. Pharmacological agents that modulate the epigenetic modifications at gene regulatory elements or directly target gene transcription and associated signaling can be used to interfere with the aberrant gene transcription observed in cohesin mutant cells.

**Figure 3 ijms-22-06788-f003:**
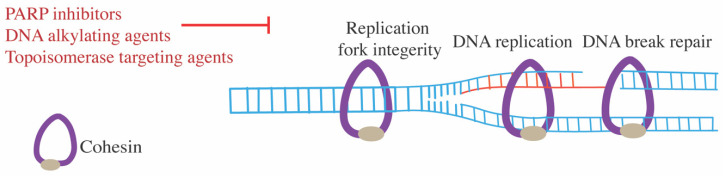
Cohesin’s role in DNA repair. Tightly regulated cohesin dynamics at replication fork ensures its integrity and restart of replication stress-induced stalled fork. Cohesin at DNA breaks facilitates homologous recombination mediated repair. Agents that disrupt DNA repair or DNA replication have been shown to have an increased inhibitory effect on cohesin-insufficient or -depleted cells.

**Table 1 ijms-22-06788-t001:** Pharmacological agents with cohesin-targeting potential.

Agent	Mode of Action	Impact on Cohesin Mutant Cells
Inactivation of STAG1	Synthetic lethal	Specific to STAG2 mutant cells [145,147,148,151].
Glycyrrhizic acid	Blocks SMC3 acetylation and interaction with RAD21 [152]	Not tested.
PCI-30451	Inhibits HDAC8 [25,146]	Not tested.
Sepin-1	Inhibits separase	Inhibits growth. Sensitises separase-overexpressing breast cancers [153,154,155].
MK-8745 ZM 44743	Inhibitors of Aurora kinase B	Differentially inhibits MCF10A cells with deletion mutations in RAD21, SMC3 and STAG2 [80].
P276-00	Inhibits cyclin-dependent kinase	Differentially inhibits MCF10A cells with deletion mutations in RAD21, SMC3 and STAG2 [80].
Decitabine Azacytidine	Hypomethylating agents	Effective in myeloid dysplasia patients with STAG2 or RAD21 mutations [109]. Differentially inhibits CD34+ cells heterozygous for SMC3 mutation [156].
JQ1	Bromodomain and extra-terminal (BET) protein inhibitor	Decreases aberrant *RUNX1* and *ERG* transcription in STAG2 mutant K562 leukaemia cells [59].
I-BET-762 RVX-208	Bromodomain and extra-terminal (BET) protein inhibitor	Differentially inhibits MCF10A cells with deletion mutations in RAD21, SMC3 and STAG2 [80].
EPZ-4777 EPZ-5676	DOTL1 inhibitors	Blocks abnormal self-renewal of mouse haematopoietic stem cells heterozygous for *Rad21* or *Smc3* mutation. Reduces aberrant HoxA7/9 expression in cohesin mutant cells [157].
LY209031	GSK3 inhibitor	Differentially inhibits MCF10A cells with deletion mutations in RAD21, SMC3 and STAG2 [80]. Differentially inhibits CMK STAG2 mutant leukaemia cells [80]. Causes enhanced β-catenin stabilization in cohesin mutant cells [80].
Lithium	GSK3 inhibitor	Rescued cell proliferation defects in Drosophila CdLS model and CdLS lymphoblastoid cells [158].
Indomethacin	Non-steroidal anti-inflammatory Wnt signalling inhibitor	Reverses the proliferation of myeloid progenitors in Nipbl mutant zebrafish [159].
WAY-600 AZD2014	mTOR inhibitor	Differentially inhibits MCF10A cells with deletion mutations in RAD21, SMC3 and STAG2 [80].
Ipatasertib	BRAF inhibitor	Differentially inhibits MCF10A cells with deletion mutations in RAD21, SMC3 and STAG2 [80].
SAR131675	VEGFR-3-tyrosine kinase	Differentially inhibits MCF10A cells with deletion mutations in RAD21, SMC3 and STAG2 [80].
VX-702	P38-MAPK/MEK inhibitor	Differentially inhibits MCF10A cells with deletion mutations in RAD21, SMC3 and STAG2 [80].
Selumetinib Trametinib	P38-MAPK/MEK inhibitors	Differentially inhibits STAG2 mutant OCI-AML3 cells [160].
Interferon	Exogenous addition of interferon	Rescues LPS-induced inflammatory response in Rad21-depleted macrophages [57].
Anti-PDL1	PDL1 inhibiton	Inhibits growth of triple-negative breast cancer cells with low Sororin and high PDL1 expression [31]. Inhibits cohesin–STAG1 function in HeLa cells [161].
Benzamide Olaparib Veliparib Rucaparib ABT-888 Talazoparib	PARP inhibitors	Differential inhibition in: Cohesin-depleted colon neoplastic cells [149]. PDS5B-depleted breast cancer cells [162]. STAG2 mutant glioblastoma, Ewing sarcoma, hTERT-positive retinal pigmented epithelial cells and leukaemia cells (U937) [15,150,163].
FK866	Nampt inhibitor. Causes hypermethylation and reduces cohesin binding in neurons [164]	Not tested.
Cyclophosphamide 5-fluorouracil	DNA alkylating agents	Differential inhibition in: RAD21-depleted MDA-MB-231 breast cancer cells [139].
Cyclophosphamide Gemcitabine Temozolomide Cisplatin	DNA alkylating agents	STAG2 mutant glioblastoma, Ewing sarcoma, hTERT-positive retinal pigmented epithelial cells [15].
VX-970 AZD6738	ATR kinase inhibitors	Differentially inhibits STAG2 mutant glioblastoma, Ewing sarcoma, hTERT-positive retinal pigmented epithelial cells [15].
Doxorubicin Etoposide Topotecan	Topoisomerase targeting agents	Differentially inhibits STAG2 mutant glioblastoma, Ewing sarcoma and hTERT-positive retinal pigmented epithelial cells [15].

## Data Availability

Not applicable.

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
