# Peer review of "Cohesin Mutations in Cancer: Emerging Therapeutic Targets"

_ijms, 2021, doi:10.3390/ijms22136788_

Round 1
Reviewer 1 Report
This manuscript by Antony et al provides a highly comprehensive review of the diverse cellular functions of Cohesins and regulatory mechanisms, especially in the context of their roles in cancers. Organization and writing is very clear and systematic. Figures are very helpful. Overall, this is an excellent review that will contribute to the field. Minor spelling checks are requested throughout- e.g. BRCA2 is listed as BRAC2; facilitates in Figure 3, etc. The sentence "The clinical prognostic impact of cohesin mutation in cancers is inclusive." is not very clear. These are very minor issues.
Reviewer 2 Report
The present manuscript is a very complete review of the “state of the art” of cohesin complex targeting in different cancers. It is very well organized and updated. I would suggest only minor changes to improve the readability of the paper.
-Formatting error on row 28
-1st paragraph it too concise and difficult to follow, I’d suggest describing better the regulatory networks mentioned and possibly to include them in the figure 1.
-Row 299-305 it is again too concise, and it is not clear what has been done
-312-313 the sentence makes no sense, why the mutual exclusivity of cohesin subunit mutations should imply that the entire complex is affected by depletion of one subunit? I think that the authors would mean that it is not possible to have multiple mutations (deletions?depletions?) in a single complex, and some subunits functions are redundant although not completely overlapping. The authors should rephrase and explain better.
The paragraph 7. “Therapeutic targeting in cohesin mutant cancers” is very interesting and useful but contains a lot of information. I would suggest recapitulating the infos in a table reporting: pathology/ mutation/synthetic lethality with pharmacological agents.
